# Association between type 1 diabetes mellitus and educational attainment in childhood: a systematic review protocol

Natalie Jayne Oakley,[1] Dylan Kneale,[2] Mala Mann,[3] Mariann Hilliar,[4] Jeanette Tan,[5] Colin Dayan,[5] John W Gregory,[1] Robert French[5]

[1]Division of Population Medicine, Cardiff University School of Medicine, Cardiff, UK
[2]Social Science Research Unit (SSRU), University College London, London, UK
[3]Specialist Unit for Review Evidence (SURE), Cardiff University School of Medicine, Neuadd Meirionnydd, University Hospital Wales, Cardiff, UK
[4]Cardiff University Library, University Hospital Wales, Heath Park, Cardiff, UK
[5]Cardiff School of Medicine, Cardiff University, University Hospital Wales, Cardiff, UK

**Correspondence to**
Dr Robert French;
frenchr3@cardiff.ac.uk

## ABSTRACT

**Introduction** Type 1 diabetes has the potential to significantly impact children's educational attainment. With the increase in incidence, quantifying this effect would be useful to assess how much additional support should be focused on children with type 1 diabetes in school.

**Methods and analysis** We will conduct a systematic review of all observational studies and randomised controlled trials, including individuals both with and without a diagnosis of type 1 diabetes who have undertaken high stakes testing at the end of compulsory schooling when under 18 years of age. The search will cover both peer-reviewed and grey literature available from January 2004 to January 2018. The following seven databases will be searched: Ovid MEDLINE (1946 to present), Ovid MEDLINE Epub Ahead of Print, In-Process & Other Non-Indexed Citations, Ovid EMBASE (1947 to present), Thomson Reuters Web of Science, EBSCO Education Resources Information Center, EBSCO British Education Index and EBSCO Cumulative Index to Nursing and Allied Health Literature. Study selection and data extraction will be performed independently by two reviewers with any disagreements resolved via a third reviewer. The quality and risk of bias in the observational studies included in this review will be assessed using the Newcastle-Ottawa Scale. We aim to conduct a meta-analysis and will assess heterogeneity between the included studies and potential for publication bias if sufficient (>10) studies are included.

**Results and dissemination** Formal ethical approval is not required as individual patient data will not be collected. Results will be disseminated through peer-reviewed publication and conference presentations.

**PROSPERO registration number** CRD42017084078.

## Strengths and limitations of this study

► This systematic review will comprehensively evaluate available literature reporting the impact of type 1 diabetes on educational attainment in individuals undertaking high stakes standardised testing under age 18 at the end of compulsory schooling.

► Our findings will be reported using the recommended methods and checklist of the Preferred Reporting Items for Systematic Reviews and Meta-Analyses.

► Study selection and data extraction will be performed independently by two reviewers, with any disagreements resolved via a third reviewer.

► A potential limitation of this review may be varying quality and high heterogeneity among available studies.

eighth edition, 451 million people aged 18–99 years worldwide are estimated to have diabetes, of which 7%–12% are thought to have T1DM. The number of children diagnosed with T1DM is increasing annually, particularly in children under 15 years of age, with an estimated annual increase of approximately 3%. In 2017, there were an estimated 587 000 children under 15 years of age with T1DM worldwide, with an estimated 96 100 new cases every year.[2] In the UK, T1DM represents over 96% of childhood cases of diabetes.[3] Short-term complications of diabetes can include hypoglycaemia, hyperglycaemia and ketoacidosis. Long-term complications include heart disease, stroke, retinopathy, neuropathy and kidney disease.[4]

### Rationale for review

The health outcomes of T1DM in children are well documented, but the wider psychosocial impacts are less established, and there is a lack of understanding of the effects on educational attainment.[5] These wider impacts are not only important in themselves, but also have the potential to have an effect on later life health outcomes through

## INTRODUCTION
### Background

Type 1 diabetes (T1DM), also known as insulin-dependent or juvenile diabetes, is an autoimmune disease which causes destruction of the insulin-producing beta cells in the pancreas, preventing the body from adequately regulating blood glucose levels. It can occur at any age but is most commonly diagnosed in childhood and adolescence.[1] According to the 2017 IDF Diabetes Atlas



mechanisms such as employment, income and social status.

Many patients and their families express concerns about the potential negative impact that T1DM may have on a child's attendance at school,[6] and many report worries about schools' ability to support children with diabetes.[7] Hypoglycaemia, hyperglycaemia and diabetic ketoacidosis as well as psychological challenges and reduced attendance due to illness and hospital appointments are all factors which may result in poorer educational attainment for children with T1DM compared with their non-diabetic counterparts.[8–10] There is conflicting evidence as to the exact effect T1DM has on educational attainment and the real magnitude of this impact.[6]

Previous literature has focused on the effects of T1DM on cognitive functioning in children. In a meta-analysis in 2008, Gaudieri et al[9] found that paediatric T1DM was found to be associated with poorer performance in learning and memory skills, as well as attention and executive function. They found that these lower cognitive scores were most pronounced with early-onset diabetes. In a further meta-analysis published in 2009 by Naguib et al,[11] T1DM in childhood was found to be associated with mild cognitive impairments and mildly reduced overall intellectual functioning. In 2004, Desrocher et al[12] published a review of the neurocognitive outcomes in children with T1DM. They reported a range of deficits associated with T1DM with most significant effects found to be related to age of disease onset, hypoglycaemia, duration of effects and hyperglycaemia around puberty. More recently in a meta-analysis in 2018, He[13] found that glycaemic extremes in children with T1DM in childhood were associated with cognitive dysfunction, characterised by lowered intelligence, reduced attention and slower psychomotor speed. These findings from previous studies suggest a detrimental impact of T1DM in childhood on cognitive function, however there is less evidence whether this adversely impacts educational attainment in the form of results of high stakes examinations.

Each of the four home nations within the UK have made a commitment to support children and young people with medical conditions in school, including T1DM. Legislation varies across the home nations but all highlight the importance of support for children and young people with additional learning needs.[14] Under the Equality Act 2010,[15] all schools in England, Scotland and Wales have a duty to make reasonable adjustments to ensure that children and young people with a disability (including T1DM) are not discriminated against or put at a significant disadvantage to their peers. In England, the Children and Families Act 2014[16] was introduced in September 2014. In January 2018, the National Assembly for Wales voted in favour of a new Additional Learning Needs and Education Tribunal Act (Wales).[17] In Scotland, there are a number of pieces of legislation regarding the rights of children with diabetes, in particular the Education (Additional Support for Learning) Act 2004 (Scotland).[18] Finally, in Northern Ireland, the Department of Education and Department of Health, Social Services and Public Safety published joint guidance entitled 'Supporting pupils with Medication Needs 2008'.[19]

As implied, both in theory and in law, T1DM has the potential to significantly impact children's educational attainment. Therefore, assessing and analysing the current evidence to quantify this effect may be useful in assessing what and how much support and educational interventions should be focused on children with T1DM in school.

## OBJECTIVES

The primary objective of this review is to assess and analyse the current literature available on whether T1DM has an impact on educational attainment in individuals undertaking high stakes standardised testing under 18 years of age at the end of compulsory schooling.

The secondary objectives include assessing the effect of T1DM on school attendance and educational attainment at other stages on the educational trajectory, if reported in the included studies.

## METHODS

We have used the Cochrane Handbook for Systematic Reviews of Interventions[20] to structure our methodological approach, and we will report our findings using the recommended methods and checklist of the Preferred Reporting Items for Systematic Reviews and Meta-Analyses (PRISMA).[21] This protocol was created using the PRISMA Protocols guidelines.[22] This protocol is registered with PROSPERO (International Prospective Register of Systematic Reviews)[23] at the NHS Centre for Reviews and Dissemination at the University of York.

### Eligibility criteria

The following criteria will be used to consider inclusion and exclusion of studies for this review.

### Type of study

We will include observational studies including prospective and retrospective cohort and case–control studies (and randomised controlled trials if available). We will exclude case series, case reports and expert opinion/narrative reviews.

### Population

We will include studies including individuals who have undertaken high stakes testing at the end of compulsory schooling when under 18 years of age.

### Intervention/exposure

Known diagnosis of T1DM before undertaking high stakes testing at the end of compulsory schooling.

### Controls/comparators

No diagnosis of T1DM before undertaking high stakes testing at the end of compulsory schooling. We will

include studies using controls which allow estimates of an interpretable effect size representative of the population, for example matched controls or population cohort controls, but excluding snowball or convenience samples. We will record the type of control in data extraction and consider the implications in the review.

### Outcome measures

The primary outcome will be grades obtained in high stakes testing at the end of compulsory schooling, taking into consideration the variation in type and timing of high stakes examinations in different countries. Secondary outcomes may include school attendance and grades obtained at other stages on the educational trajectory if reported in included studies.

### Time frame

The 2015 National Institute for Health and Care Excellence guidelines state that since 2004 there have been major changes in routine management of T1DM, aiming to achieve better glucose control to reduce long-term complications associated with the condition.[24] We will, therefore, include studies published after the year 2004 in order to comprehensively evaluate the most up-to-date available peer-reviewed and grey literature. The effect on educational attainment associated specifically with these treatment changes from 2004 may only become apparent at a later stage and therefore only seen in more recent or future studies. As a result, while it is likely that many qualifying studies will use cohorts receiving treatment prior to this year, we will record this as part of our data extraction and consider this as part of the review comparison.

### Setting

Included studies will be secondary school based. Studies including outcomes from educational tests undertaken in clinical or other non-school settings will be excluded.

### Search methods for identification of studies

We will search the following databases from January 2004 to January 2018 and will consider only studies published in English.

- ► Ovid MEDLINE (1946 to present).
- ► Ovid MEDLINE Epub Ahead of Print, In-Process & Other Non-Indexed Citations.
- ► Ovid EMBASE (1947 to present).
- ► Thomson Reuters Web of Science.
- ► EBSCO Education Resources Information Center.
- ► EBSCO British Education Index.
- ► EBSCO Cumulative Index to Nursing and Allied Health Literature.

Comprehensive electronic literature search strategies will be used for each database. See online supplementary appendix 1 for the Ovid MEDLINE and Ovid EMBASE search strategy.

To identify additional papers, information on studies in progress, unpublished research or research reported in the grey literature will be identified through searching a range of relevant websites, including diabetes.org.uk, and trial registers including clinicaltrials.gov. We will search Electronic Table of Contents of key journals for relevant studies that have been published within the last 2 years. We also plan to check review articles, reference lists and carry out citation tracking of included studies for any significant studies missed during the database search.

### Selection of studies

To select studies for further assessment, they will be imported and organised into Eppi-Reviewer V.4.0[25] and duplicates will be removed. Two independent reviewers (NO and RF) will screen the titles and abstracts of records retrieved from the searches using the predetermined inclusion criteria using Eppi-Reviewer V.4.0.[25] Records identified as potentially eligible on the basis of title and abstract will then be screened on full text according to set inclusion criteria. If there is any doubt or disagreement regarding study selection, there will be further discussion and, if required, involvement of a third reviewer (JG) to reach a consensus. Rationale for exclusion of studies at this stage will be documented. The remaining included studies will then undergo data extraction using a standardised pro forma. A PRISMA flow diagram will be used to demonstrate the number of included and excluded studies.

### Data collection

All included studies will undergo data extraction by two independent reviewers (NO and RF), using a standardised pro forma. The pro forma will be pilot tested initially to ensure consistency.

Data extracted from each study will include:

- ► Details of study, for example, first author, date of publication, country/region where study was undertaken.
- ► Details of study methodology, for example, study design, sample size, number of cases and controls included, inclusion/exclusion criteria, data linkage.
- ► Modelling strategy and covariates/confounders adjusted for, for example, age, gender, socioeconomic group, age at diagnosis, duration of diabetes.
- ► Outcomes—as stated below.

Again, any disagreements will be discussed and a third reviewer (JG) will be consulted if required.

### Outcomes and prioritisation

#### Primary outcome

The primary outcome will be grades obtained in high stakes testing at the end of compulsory schooling. In most cases, we expect this to be a continuous measure assessing scores across a range of subjects. We anticipate there may be some cases where a binary measure is used, for example, achieving five General Certificates of Secondary Education (GCSEs) (grades A*–C) is a commonly used benchmark in UK educational research.

#### Secondary outcomes

The secondary objectives may include school attendance and grades obtained at other stages on the educational

trajectory if reported. Again, in most cases, we expect these to be continuous measures.

### Missing data

For any questions about eligibility or data not obtained from the full paper review, the authors of the papers will be contacted if required. If after 6 weeks no clarification has been provided, the study will be included in the final analysis and discussion, however will be identified as ideally requiring further information.

### Assessment of risk of bias in included studies

The quality and risk of bias in the observational studies included in this review will be assessed using the Newcastle-Ottawa Scale (NOS) for assessing the quality of non-randomised studies in meta-analysis.[26] The NOS assesses cohort and case–control studies based on three domains:

1. Selection of study groups.
2. Comparability of study groups.
3. Ascertainment of exposure (case–control studies)/ outcome (cohort studies).

Each study can be awarded a maximum of one star for each numbered component within the selection and exposure sections and a maximum of two stars can be given for the comparability section, creating a maximum of nine stars per study. The higher the number of stars, the better quality the study and the lower the risk of bias.

If any randomised controlled trials are identified for inclusion in this review, we will assess the quality and risk of bias using the Cochrane Risk of Bias Tool, as described in the Cochrane Handbook for Systematic Reviews of Interventions.[27] This tool assesses risk of bias using five main domains: selection bias, performance bias, reporting bias, detection bias and attrition bias. It allows categorisation of risk of bias using three main outcomes: high, low or unclear.

We will also specifically analyse the linkage methodology used in all papers included, highlighting areas of potential bias which may impact on the overall quality of the studies.

In our review, this assessment will be completed by two independent reviewers (NO and RF). Any disagreements that cannot be resolved during moderation will be discussed with a third reviewer (JG).

### Data synthesis

We will aim to conduct a meta-analysis using a random-effects model.

The majority of the outcome data from included studies in our review is likely to be continuous, therefore the measure of effect will be analysed using standardised mean difference with 95% CI. Any dichotomous outcome data will be analysed using risk ratios or ORs which will also be converted into standardised mean difference with the appropriate transformations.

In order to not lose information, we will convert measures into a common metric and will aim to undertake sensitivity analyses to look for systematic difference according to transformations. We will use the statistical software Eppi-Reviewer V.4.0[25] for our meta-analysis.

If possible, a sensitivity analysis will be performed to explore the impact of decisions made during the calculation of effect sizes, the inclusion of different study designs and the impact of risk of bias assessments.

If we are unable to analyse data using meta-analysis, we will conduct a narrative synthesis. In this case, we will narratively summarise and tabulate the results found during data extraction in order to identify patterns in study design and outcomes across the included studies.

### Assessment of heterogeneity

We will assess heterogeneity between the included studies by visual assessment of forest plots (for any minimal overlap) and use of statistical tests including the $\chi^2$ test and the $I^2$ statistic. If there is evidence of statistical heterogeneity, we will attempt to explore the reasons for the heterogeneity by using subgroup analyses based on the following:

▶ Patient demographics, for example, age, gender.
▶ Diabetes-specific characteristics, for example, age at diagnosis, haemoglobin A1c.

We will also consider a random-effects meta-regression.

### Publication bias

We will examine funnel plots and conduct tests (Egger's test) to assess the potential for publication bias where there are sufficient (>10) studies.

### Quality of overall body of evidence

We will assess the quality of evidence for all outcomes using the Grading of Recommendations Assessment, Development and Evaluation system. Risk of bias, directness, precision, heterogeneity and publication bias will be assessed, and quality of the evidence will then be judged as high, moderate, low or very low. Results will be presented in 'Summary of findings' tables as recommended by the Cochrane Handbook for Systematic Reviews of Interventions.[27]

### Patient and public involvement

Patients were not involved in the development of this research question or systematic review protocol. Patients will not be involved in completion of the systematic review.

**Contributors** RF is the review guarantor. The concept of the review was proposed by RF and JWG, and the protocol manuscript was drafted by NJO and edited by RF, MM and JWG. The search strategy was designed by MH, NJO and RF with advice from MM. NJO, MM, JT and RF contributed to the development of the study eligibility criteria and data extraction criteria. JWG and CD provided expertise on type 1 diabetes. MM provided expertise on systematic review methodology. DK provided expertise on data extraction and meta-analysis. All authors read, edited and approved the final manuscript.

**Funding** This research was supported in part by the MRC grant MR/N015428/1.

**Competing interests** RF has received a grant from the Medical Research Council MR/N015428/1 for his work as principal investigator of the project 'Investigating the inter-relationship between diabetes and children's educational achievement'.

**Patient consent** Not required.

**Provenance and peer review** Not commissioned; externally peer reviewed.

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
