## [Reviewer comments · BMJ Open]

ARTICLE DETAILS

TITLE (PROVISIONAL)	The Association between Type 1 Diabetes Mellitus and Educational Attainment in Childhood: A Systematic Review Protocol
AUTHORS	Oakley, Natalie; Kneale, Dylan; Mann, Mala; Hilliar, Mariann; Tan, Jeanette; Dayan, Colin; Gregory, John; French, Robert

VERSION 1 – REVIEW

REVIEWER	Malin Rising Holmström Department of Nursing Mid Sweden University Holmgatan 10, SE-851 70 Sundsvall, Sweden, Mobile +46(0)10 142 84 23
REVIEW RETURNED	03-May-2018

GENERAL COMMENTS	A very interesting and important review that will add new valuable knowledge to the area. However some of the references could be more up-to -date (Matyka et al). Furthermore I wonder regarding the methods section. //Population: We will include studies including individuals who have undertaken high stakes testing at the end of compulsory schooling when under 18 years of age. Intervention/Exposure: Known diagnosis of type 1 diabetes before undertaking high stakes testing at the end of compulsory schooling. Controls/Comparators: No diagnosis of type 1 diabetes before undertaking high stakes testing at the end of compulsory schooling.// This control group is likely to contain youth with a various number of diagnoses (not diabetes) but, for example, ADHD, depending on the school system where the studies are conducted. What are the thoughts from the authors how to manage this potential bias?
---

REVIEWER	Matt Cooper Telethon Kids Institute, Australia
REVIEW RETURNED	07-Jun-2018

GENERAL COMMENTS	The question being asked in this systematic review is a worthy academic pursuit, it is not an easy area of the literature to navigate with sparse studies and different outcomes used across the studies that have been carried out. The approach, in general, is sound, however, the protocol is very light on details, both at the background level and at the study design level. It is challenging to give specific advice in this case (for a protocol) compare to a completed study write up, however, the following points should be considered during revision. Major comments:
---

	- Background is very brief, and should at least acknowledge the literature around cognitive function testing in the rationale for a possible impact of T1D on educational attainment - 'As a result, we will include studies published after the year 2004.' Is this the correct approach? Presumably a study published in 2005 will use data collected between 1995 and 2001 (as an example) from this time of 'different treatment', how will this affect your study? - Duration and/or age of T1D diagnosis are likely to play a significant role in the relationship between T1D status and educational attainment, yet they aren't mentioned in the protocol/analysis plan Specific minor comments: - 'school attainment' used in the 2nd sentence of abstract, as opposed to 'educational attainment' (title) - strengths statement mentions 'educational achievement', rationale mentions 'educational experience' the language is inconsistent - '... how much support and interventions should ...' abstract, revise grammar - 'and potential for publication bias if sufficient studies.' abstract, incomplete sentence? - background 'Although in the population 10 per cent of people with diabetes have type 1,(4) it represents over 98% of childhood cases of diabetes.' Per cent and % used, consistency. In general this sentence could be revised. - 'Many patients and their families express concerns about the potential negative impact that T1DM may have on a child's attainment at school.' – is there a reference for this? - The statement 'Currently laws relating to managing children with chronic disease in school in the UK vary depending on specific country.' – does not seem supported by the following text, as only information about England is presented. - 'We will search the following databases from 2004 to present' – the abstract states Jan 2004 to Jan 2018, revise
--	---

VERSION 1 – AUTHOR RESPONSE

Section 1: Malin Holmström's (Reviewer 1) comments

'A very interesting and important review that will add new valuable knowledge to the area. However some of the references could be more up-to -date (Matyka et al).'

We have subsequently edited the Introduction sections 'Background' and 'Rationale for Review' on pages 2 and 3, making sure to include more up to date references, as seen on pages 8 and 9.:

New reference for T1DM definition:

1. Krzewska A, Ben-Skowronek I. Effect of associated autoimmune diseases on type 1 diabetes mellitus incidence and metabolic control in children and adolescents. *BioMed Research International*. 2016; 2016: 6219730.

New reference for international prevalence:

2. International Diabetes Federation. *IDF Diabetes Atlas*, 8th edn. Brussels, Belgium: International Diabetes Federation, 2017.

New reference for UK prevalence:

3. Royal College of Paediatric and Child Health. *National Paediatric Diabetes Audit 2013-14. Report 1: Care Processes and Outcomes*. Royal College of Paediatrics and Child Health; London (UK): Revised Sept 2015

New reference for statement that wider psycho-social impacts of T1D not well documented

5. Persson S, Dahlquist G, Gerdtham UG and Steen Carlsson K. Impact of childhood-onset type 1 diabetes on schooling: a population-based register study. *Diabetologia* 2013; 56: 1254-1262.

New reference for statement that families report concerns of the support in schools for children with T1DM:

7. Streisand R, Monaghan M. Young Children with Type 1 Diabetes: Challenges, Research, and Future Directions. *Current diabetes reports*. 2014;14(9):520. doi:10.1007/s11892-014-0520-2.

Additional references for effect of T1D on cognitive outcomes (12-14):

12. Desrocher M, Rovet J. Neurocognitive correlates of type 1 diabetes mellitus in childhood. *Child Neuropsychol* 2004; 10(1): 36-52.

13. He J et al. Glycaemic extremes are related to cognitive dysfunction in children with type 1 diabetes: a met-analysis. *J Diabetes Investig*. 2018. doi: 10.1111/jdi.12840. [Epub ahead of print]

14. Naguib et al. Neuro-cognitive performance in children with type 1 diabetes – a meta-analysis. *J Pediatr Psychol*. 2009; 34(3): 271-82.

Additional references for the legal obligations for support for children with T1D in UK schools (16-20):

16. Equality Act 2010 (England, Scotland, Wales). [Available from http://www.legislation.gov.uk/ukpga/2010/15/pdfs/ukpga_20100015_en.pdf] (Accessed 12th June 2018)

17. Children and Families Act 2014 (England). [Available from http://www.legislation.gov.uk/ukpga/2014/6/pdfs/ukpga_20140006_en.pdf] (Accessed 12th June 2018)

18. Additional Learning Needs and Education Tribunal Act 2018 (Wales). [Available from http://www.legislation.gov.uk/anaw/2018/2/pdfs/anaw_20180002_en.pdf] (Accessed 12th June 2018)

19. Education (Additional Support for Learning) Act 2004 (Scotland). [Available from https://www.legislation.gov.uk/asp/2004/4/pdfs/asp_20040004_en.pdf] (Accessed 12th June 2018)

20. Department of Education and Department of Health, Social Services and Public Safety. Supporting pupils with Medication Needs 2008. [Available from: <https://www.education-ni.gov.uk/sites/default/files/publications/de/supporting-pupils-with-medical-needs.pdf>] (Accessed 12th June 2018)

Furthermore I wonder regarding the methods section.

//Population: We will include studies including individuals who have undertaken high stakes testing at the end of compulsory schooling when under 18 years of age.

Intervention/Exposure: Known diagnosis of type 1 diabetes before undertaking high stakes testing at the end of compulsory schooling.

Controls/Comparators: No diagnosis of type 1 diabetes before undertaking high stakes testing at the end of compulsory schooling.//

This control group is likely to contain youth with a various number of diagnoses (not diabetes) but, for example, ADHD, depending on the school system where the studies are conducted.

What are the thoughts from the authors how to manage this potential bias?'

This is a very important consideration, we have extended the 'Controls/Comparators' section on page 4 to state that we will include this as part of our bias evaluation. The new text is pasted below:

Controls/Comparators:

No diagnosis of type 1 diabetes before undertaking high stakes testing at the end of compulsory schooling. **We will include studies using controls which allow estimates of an interpretable effect size, for example matched controls or population controls. We will record the type of control in data extraction and consider the implications in the review.**

- We will include studies using controls which allow estimates of an interpretable effect size, for example matched controls or population controls. We will record the type of control in data extraction and consider the implications in the bias section of the review.

Section 2: Matt Cooper's (Reviewer 2) comments:

Major comments:

'Background is very brief, and should at least acknowledge the literature around cognitive function testing in the rationale for a possible impact of T1D on educational attainment'.

- We have edited the Introduction sections 'Background' and 'Rationale for Review' on pages 2 and 3, making sure to include more information about previous literature regarding the effect of type 1 diabetes in childhood on results of cognitive function testing.

'As a result, we will include studies published after the year 2004.' Is this the correct approach? Presumably a study published in 2005 will use data collected between 1995 and 2001 (as an example) from this time of 'different treatment', how will this affect your study?

Duration and/or age of T1D diagnosis are likely to play a significant role in the relationship between T1D status and educational attainment, yet they aren't mentioned in the protocol/analysis plan'

- We found the comment regarding year of publication of included studies particularly interesting and thought-provoking, and as a result have included further detail regarding the rationale for this in the Methods 'Time frame' section on pages 4 and 5. We will include studies published after the year 2004 in order to comprehensively evaluate the most up-to-date available peer-reviewed and grey literature. We note that, according to the 2015 NICE guidelines, there have been major changes in routine management of type 1 diabetes since 2004. We acknowledge that the effect on educational attainment associated specifically with these treatment changes may only become apparent at a later stage and therefore only seen in recent or future studies. As a result, while it is likely that many qualifying studies will use cohorts receiving treatment prior to this year, we will record this as part of our data extraction and consider this as part of the review comparison. New text pasted below:

Time frame:

The 2015 NICE guidelines state that since 2004 there have been major changes in routine management of type 1 diabetes, aiming to achieve better glucose control to reduce long term complications associated with the condition.(24) **We will therefore include studies published after the year 2004 in order to comprehensively evaluate the most up-to-date available peer-reviewed and grey literature. The effect on educational attainment associated specifically with these treatment changes from 2004 may only become apparent at a later stage and therefore only seen in more recent or future studies. As a result, while it is likely that many qualifying studies will use cohorts receiving treatment prior to this year, we will record this as part of our data extraction and consider this as part of the review comparison.**

- We have added 'age at diagnosis' and 'duration of diabetes' into the data to be extracted from each study in the 'Data collection' section on page 5. This information will therefore be collected when available and used as part of the review comparison.

Specific minor comments:

- 'school attainment' used in the 2nd sentence of abstract, as opposed to 'educational attainment' (title)

- Language amended accordingly in the abstract introduction on page 1.

- strengths statement mentions 'educational achievement', rationale mentions 'educational experience' the language is inconsistent

- Language changed to 'educational attainment' throughout manuscript to improve consistency, as can be seen in abstract introduction on page 1, strength and limitations of this study on page 2, rationale for review on page 3.

- '... how much support and interventions should ...' abstract, revise grammar

- Sentence edited and grammar amended accordingly in the abstract introduction on page 1.
- 'and potential for publication bias if sufficient studies.' abstract, incomplete sentence?
 - Sentence amended accordingly in 'Methods and analysis' section of abstract on page 1.
- background 'Although in the population 10 per cent of people with diabetes have type 1,(4) it represents over 98% of childhood cases of diabetes.' Per cent and % used, consistency. In general this sentence could be revised.
 - Paragraph edited, statistics updated and language changed to 'percent' throughout Introduction 'Background' section on page 2 to improve consistency.
 - - 'Many patients and their families express concerns about the potential negative impact that T1DM may have on a child's attainment at school.' – is there a reference for this?
 - Sentence edited and references amended accordingly in Introduction 'Rationale for Review' section on pages 2 and 3, and references on page 8 and 9.
- The statement 'Currently laws relating to managing children with chronic disease in school in the UK vary depending on specific country.' – does not seem supported by the following text, as only information about England is presented.
 - Paragraph amended in Introduction 'Rationale for Review' section on page 3. Information added regarding legislation throughout the UK - in Scotland, Wales and Northern Ireland as well as England.
- 'We will search the following databases from 2004 to present' – the abstract states Jan 2004 to Jan 2018, revise'
 - Date amended accordingly to 'January 2004 to January 2018' in Search methods for identification of studies' section on page 5.

Section 3: Editors amendments

Finally, the following formatting amendments have been made to the manuscript as requested:

FORMATTING AMENDMENTS FROM EDITORIAL OFFICE:

- Please re-upload APPENDIX in PDF format.

- Uploaded accordingly as supplementary file.

- We have implemented an additional requirement to all articles to include 'Patient and Public Involvement statement' within the main text of your main document. Authors must include a statement in the methods section of the manuscript under the sub-heading 'Patient and Public Involvement'. This should provide a brief response to the following questions:

How was the development of the research question and outcome measures informed by patients' priorities, experience, and preferences?

How did you involve patients in the design of this study?

Were patients involved in the recruitment to and conduct of the study?

How will the results be disseminated to study participants?

For randomised controlled trials, was the burden of the intervention assessed by patients themselves?

Patient advisers should also be thanked in the contributorship statement/acknowledgements.

If patients were not involved please state this.

'Patient and Public Involvement' statement included on page 7.

VERSION 2 – REVIEW

REVIEWER	Malin Rising Holmström Universitetslektor / Senior lecturer Avdelningen för omvårdnad / Department of Nursing Mittuniversitetet / Mid Sweden University, Holmgatan 10, SE-851 70 Sundsvall, Sweden, Mobile +46(0)10 142 84 23
REVIEW RETURNED	17-Jul-2018
GENERAL COMMENTS	Thank you for a very good reply from the authors and also good improvements made in the paper. Good luck in future research!
REVIEWER	Matthew Cooper Telethon Kids Institute, Australia, The University of Western Australia, Australia
REVIEW RETURNED	31-Jul-2018
GENERAL COMMENTS	The revisions have satisfactorily resolved in the queries I raised. I encourage the researchers to continue to think about the possible temporal elements of this study, and how it may impact the analysis, as they work through extracting/assessing the papers.